# Investigating the potential of the secretome of mesenchymal stem cells derived from sickle cell disease patients

**Tiago O. Ribeiro**[1]ᐤ, **Brysa M. Silveira**[1]ᐤ, **Mercia C. Meira**[1], **Ana C. O. Carreira**[2], **Mari Cleide Sogayar**[2,3], **Roberto Meyer**[1], **Vitor Fortuna**[1]*

**1** Health Science Institute, Federal University of Bahia, Salvador, BA, Brazil, **2** Cell and Molecular Therapy Center NUCEL-NETCEM, School of Medicine, Internal Medicine Department, University of São Paulo, São Paulo, SP, Brazil, **3** Chemistry Institute, Biochemistry Department, University of São Paulo, São Paulo, SP, Brazil

ᐤ These authors contributed equally to this work.

* vfort@ufba.br

**Data Availability Statement:** The authors declare that all data underlying the findings are fully

## Abstract

Sickle cell disease (SCD) is a monogenic red cell disorder associated with multiple vascular complications, microvessel injury and wound-healing deficiency. Although stem cell transplantation with bone marrow-derived mesenchymal stem cells (BMSC) can promote wound healing and tissue repair in SCD patients, therapeutic efficacy is largely dependent on the paracrine activity of the implanted BM stromal cells. Since *in vitro* expansion and culture conditions are known to modulate the innate characteristics of BMSCs, the present study investigated the effects of normoxic and hypoxic cell-culture preconditioning on the BMSC secretome, in addition to the expression of paracrine molecules that induce angiogenesis and skin regeneration. BMSCs derived from SCD patients were submitted to culturing under normoxic (norCM) and hypoxic (hypoCM) conditions. We found that hypoxically conditioned cells presented increased expression and secretion of several well-characterized trophic growth factors (VEGF, IL8, MCP-1, ANG) directly linked to angiogenesis and tissue repair. The hypoCM secretome presented stronger angiogenic potential than norCM, both *in vitro* and *in vivo*, as evidenced by HUVEC proliferation, survival, migration, sprouting formation and *in vivo* angiogenesis. After local application in a murine wound-healing model, HypoCM showed significantly improved wound closure, as well as enhanced neovascularization in comparison to untreated controls. In sum, the secretome of hypoxia-preconditioned BMSC has increased expression of trophic factors involved in angiogenesis and skin regeneration. Considering that these preconditioned media are easily obtainable, this strategy represents an alternative to stem cell transplantation and could form the basis of novel therapies for vascular regeneration and wound healing in individuals with sickle cell disease.

## Introduction

Sickle cell disease (SCD), the most common inherited hemoglobinopathy worldwide, is characterized by repeated vaso-occlusion crises secondary to sickled red blood cells [1]. It is associated with significant microvessel injury, as well as impairments in neovascularization, wound

available without restriction. All relevant data are within the Supporting Information files.

**Funding:** This work was financially supported by the Brazilian Ministry of Health, the Brazilian National Research Council (CNPq) grant 443137/2016-1 to VF, and Research Support Foundation of the State of Bahia (FAPESB). The funders had no role in study design, data collection and analysis, decision to publish, or preparation of the manuscript.

**Competing interests:** The authors have declared that no competing interests exist.

**Abbreviations:** ANG, Angiogenin; BMSC, Bone marrow–derived mesenchymal stromal/stem cell; CM, Conditioned medium; COL1A1, Collagen type I alpha 1 chain; COL4, Collagen type IV alpha 4 chain; DAPI, 4′,6-diamidino-2-phenylindole; DMEM, Dulbecco's modified Eagle's medium; EG-VEGF, Endocrine gland-derived vascular endothelial growth factor; EPO, Erythropoietin; EPC, Endothelial progenitors cell; FBS, Fetal bovine serum; FGF1, Fibroblast growth factor 1; GAPDH, Glyceraldehyde 3-phosphate dehydrogenase; HbS, hemoglobin S; HGF, Hepatocyte growth factor; HPRT1, hypoxanthine phosphoribosyltransferase 1; HUVEC, Human umbilical vein endothelial cells; HypoCM, Hypoxic conditioning medium; IL8, Interleukin 8; IGFBP-1/2/3, Insulin-like growth factor-binding protein 1/2/3; MCP-1, Monocyte chemoattractant protein-1; MMP2, Matrix metallopeptidase 2; NG2, Neural/glial antigen 2 proteoglycan; NorCM, Normoxic conditioning medium; PBS, Phosphate buffered solution; PCOLCE, Procollagen C-endopeptidase enhancer; PDGFR-B, Platelet-derived growth factor receptor beta; PIGF, Placental growth factor; PTX3, Pentraxin-related protein 3; RN18S1, RNA, 18S ribosomal 5; SCD, Sickle cell disease; SDF1a, Stromal cell-derived factor-1; SPARC, Secreted protein acidic and cysteine rich; TGFB1, Transforming growth factor beta-induced; THBS1, Thrombospondin 1; TIMP-4, Metalloproteinase inhibitor 4; VEGF-A, Vascular endothelial growth factor A.

healing and tissue repair [2,3]. SCD patients are at high risk of a wide range of complex and multifactorial vasculopathic complications, including pulmonary hypertension, retinopathy, priapism, osteonecrosis and leg ulcers [4,5]. Consequently, these complications often cause significant functional, emotional, and economic burdens for the afflicted patients and result in considerable cost to the healthcare system [6, 7].

The transplantation of bone marrow-derived mesenchymal stem cells (BMSC) has been extensively investigated as source of promising proangiogenic stem cell therapy for diseases with vascular complications, such as peripheral artery disease, acute kidney injury, myocardial infarction and skin ulcers [8]. A growing number of studies have reported that BMSC secrete a wide range of bioactive factors that enhance *in vitro* the proliferation and migration of endothelial cells [9, 10] and promote tissue healing and formation of new blood vessels [11]. Recently, Kim and colleagues identified important bioactive factors in the BMSC secretome that correlate with vascular regenerative efficacy in the treatment of ischemic disease [10]. These biofactors were then validated and can now be used as efficient biomarkers to predict response to proangiogenic MSC-based cell therapies. Furthermore, significant variation in the MSC secretome and the functional capacity of its biomarkers has been observed among differing donor sources and diseases. However, in SCD, the key factors secreted by BMSCs that possess the potential to promote angiogenesis and tissue repair have not been identified to date.

As cell therapy efficacy is dependent on the number of implanted BMSCs, culture expansion can overcome this limitation to improve the treatment of diseases with vascular complications [12]. However, *in vitro* expansion and culture conditions modulate the innate characteristics of BMSCs and hinder the clinical applications of BMSCs [13, 14]. To optimize the culturing conditions of stem cells, various *in vitro* pretreatment strategies ("preconditioning") have recently been evaluated to enhance the regenerative capacity of BMSCs, including cell culture expansion in an hypoxic (Hyp) environment [15]. Preconditioning by hypoxia increases the secretion of regenerative factors and enhances stem cell survival [16, 17]. The paracrine factors secreted by cells can accumulate in the conditioned medium (CM). The conditioned medium derived from the BMSC culture has been reported to serve multiple positive functions in tissue regeneration [11, 12, 16, 18]. Furthermore, findings by Elabd and colleagues suggest that hypoxic preincubation positively impacted the BMSC secretome and transcriptome, improving the vasculogenic and angiogenic properties critical for the development of successful cellular therapies [19]. Although numerous studies using BMSCs and their conditioned mediums as potential therapeutic agents have been published [18, 20–22], how hypoxic preincubation affects the BMSC secretion of bioactive factors with vascular regenerative potential remains poorly understood.

Here, we attempted to investigate the potential of BMSCs from SCD patients as an innovative source for proangiogenic therapies. We first evaluated the effects of hypoxic preconditioning on the ability of these BMSCs to secrete bioactive angiogenic factors in culture medium (CM). We then used this hypoxic preconditioned CM for subsequent *in vitro* and *in vivo* studies in a mouse model of angiogenesis and wound healing. In summary, preconditioning by hypoxia was shown to enhance the secretion of trophic paracrine factors involved in angiogenesis and tissue regeneration. Therefore, hypoxic BMSCs secretome represent a promising alternative for stem cell transplantation and can provide a novel proangiogenic and tissue regeneration option for sickle cell disease patients.

## Material and methods

### Cell isolation and culturing

Human umbilical vein endothelial cells (HUVEC) were isolated as described previously [23]. The institutional review board of the Climério de Oliveira Maternity Hospital (Federal

University of Bahia, approval n° 625.059) approved the human umbilical cord sampling and bone marrow aspiration procedures after obtaining written informed patient consent. Briefly, the umbilical vein was cannulated with a 20G needle and perfused with 0.05% collagenase I at 37˚C for 15 min. HUVECs were then collected with PBS, centrifuged at 200 g for 10 min and resuspended in EGM2/ BulletKit medium (Lonza Group Ltd.). HUVECs were cultured on 0.1% gelatin (Sigma, St. Louis, MO) in EGM2/BulletKit supplemented with 100 U/mL penicillin/streptomycin (Life Technologies) at 37˚C in 5% $CO_2$ and 95% air. The medium was replaced every 2–3 days and cells were subcultured when confluent. The HUVECs were then maintained in EGM2 and used for experiments for a maximum of five passages.

BM was aspirated from the posterior iliac crest of nine patients with SCD. BMSC were cultured as previously described [24]. Briefly, BM aspirates were diluted with PBS, layered onto a Ficoll density gradient and centrifuged at 400 g for 20 minutes. Mononuclear cells were plated in complete Dulbecco's modified Eagle's complete medium (low glucose, 10% fetal calf serum, 100 U/mL penicillin/streptomycin) at 100,000 to 300,000 cell/cm$^2$. After four days of culturing, the medium was replaced and BMSCs were allowed to expand for 7–12 days. BMSCs were passaged weekly, and passages 3–6 were used in experimentation.

## Flow cytometric analysis of BMSCs from SCD patients

BMSCs were harvested after the third passage and cell surface antigen expression was analyzed using a fluorescence-activated cell sorter (FACScalibur, BD Biosciences). The following monoclonal antibodies were used: CD29-FITC (clone TS2/16), CD90-FITC (clone eBIO5E10), CD105-PE (clone SN6), CD14-PE (clone 61D3) (all from eBioscience, San Diego, CA, USA), CD31-FITC (clone WM59), CD34-FITC (clone 8G12) (both from BD Biosciences, San Jose, CA, USA) and CD73-PE (clone AD2, EXBio Praha). All samples were run using the appropriate isotype control antibody in accordance with the manufacturer's protocol.

## Preconditioning and conditioned medium (CM) collection

BMSCs were expanded to subconfluency in standard culture medium at 37˚C under 5% $CO_2$ and humidity (standard conditions). To generate normoxic (nor) or hypoxic (hypo) conditions, confluent cells in 6-well plates (average of 20,000 cells/cm$^2$) were washed twice and cultured for 48h with 3ml/well of serum-free EBM-2 medium. For hypoxic preconditioning, BMSC were placed in an Anaerobac Jar (Probac, São Paulo, Brazil) at 0.5% oxygen, as described [17, 25]. After 48hs, the procedures for conditioned medium collection and storage were performed as previously described [26]. Briefly, the cell-cultured supernatant was aspirated, pooled, centrifuged at 2000g for 20 min at 4° C, and stored at -70° C. This total BMSC secretome hereafter is referred to as "BMSC secretome". Total protein was determined by the Lowry method before use.

## Proteome profiler array studies

Human angiogenesis antibody arrays were purchased from R&D Systems (cat. n° ARY007) and used according to the manufacturer's instructions. Briefly, protein contents in the CM were determined by Lowry assay and equal amounts of protein were used for each condition. After overnight incubation, retained protein signals were revealed by an enhanced luminol-based detection reaction and quantified by densitometry using ImageJ (NIH, USA) software.

## Analysis of mRNA expression by real-time qPCR

After normoxic or hypoxic conditions, BMSC monolayers were briefly rinsed with ice-cold PBS and total RNA was extracted using a commercial kit (RNeasy Qiagen) according to the

manufacturer's instructions. All RNA samples were checked for purity using Thermo Scientific NanoDrop spectrophotometer. 1 μg RNA was taken for cDNA synthesis with the Omniscript RT-PCR kit from Qiagen as recommended by the manufacturer. Relative mRNA expression of the target genes was detected using Maxima SYBR Green/ROX qPCR Master Mix (Thermo Scientific) (final volume: 12 μL) in a GeneAmp 7300 Sequence Detection System (Applied Biosystems) in accordance with the manufacturer's instructions. All primer sequences are listed in a S1 Table. Differential mRNA expression levels were considered when relative expression (R.E.) was ≥2.0 and ≤0.5 and statistical significance was determined if $P < 0.05$ (Wilcoxon's test). Relative mRNA expression levels were estimated using the method described by [27], which generates normalized values in geNorm Software using *RN18S1*, *GAPDH* and *HPRT* for housekeeping.

## Proliferation and TUNEL assays

HUVECs were seeded (2.0 x $10^4$ cells/well) on 0.1% gelatin-coated 24-well plates and serum starved for 4 hours in EBM-2 medium, followed by stimulation with norCM, hypoCM or a vehicle control for 24 hours. Cultured HUVECs were incubated with 10 μM BrdU (5-bromo-2′-deoxy-uridine) for 4 h before fixation. BrdU detection and incorporation were performed with a cell proliferation kit (Thermo Fisher Scientific).

Fluorescent terminal deoxynucleotidyl transferase deoxyuridine triphosphate (dUTP) nick end labeling (TUNEL) was performed using an in situ cell death detection kit (Roche) in accordance with the manufacturer protocol. Cell death in HUVEC was induced with serum starvation for 4 h. Only apoptotic cells were stained by TUNEL, while DAPI labeled the nuclei of all cells. The number of HUVECs that underwent apoptosis was determined by counting TUNEL/DAPI double-stained cells in five different fields under a fluorescence microscope.

## Bead-sprouting angiogenesis assay

A 3-dimensional *in vitro* model of angiogenesis was utilized, as previously described [28, 29]. Briefly, collagen-cytodex microcarrier beads (Sigma-Aldrich Co) were coated with HUVECs (#3) and embedded in fibrin gel. After the gels were allowed to clot on 24-well plates, vehicle control, nor- or hypo-BMSC-derived conditioned medium (500 μL) was added to each well and replaced every two days. The number of endothelial sprouts/beads, branches and tubule length was quantified in at least 30 beads per condition after four days. Only sprouts larger than the bead diameter were considered.

## *In vivo* angiogenesis assay

All procedures were approved by the institutional review board for animal experimentation (CEUA-2018-131). *In vivo* angiogenesis experiments were performed as previously described [30]. Briefly, a mixture of basement membrane matrix (Geltrex reduced growth factor, Gibco) and conditioned medium (9:1 proportion) was prepared in the 20-mm sterile surgical silicone tubes (angioreactors) from a DIVAA kit (R&D system). A mixture containing basement membrane matrix and buffered saline was included as a control during the assay. After a 1-hour incubation period at 37° C, the angioreactors were subcutaneously implanted into the dorsal flanks of 8-week-old female C57/BL6 mice. Up to 2 angioreactors were implanted on each side for a total of 4 angioreactors per mouse (n = 4 animals per group, for each independent experiment). After 11 days, the animals were euthanized and the angioreactors were removed. For quantitation of functional vessels formed, the contents were removed from angioreactors and homogenized in distilled water. The relative hemoglobin content was measured at 540 nm according to [10]. Differences between the conditions were compared using the Student's t-test.

## Scratch wound assay

HUVEC monolayers cultured on 24-well plates were scratched with a pipette tip to create a straight gap. After debris removal via extensive washing, cells were allowed to migrate in the presence of norCM, hypoCM or a vehicle control medium for 24 hours. In order to evaluate the same field during image acquisition using a phase-contrast microscope, reference points were marked close to the scratches. The plates were kept in an incubator at 37°C and examined periodically (0, 6, 12 and 24 h). The open wound area visible in each image was quantitatively measured at each time point using ImageJ program software.

## Gel electrophoresis and western blot analysis

HUVECs were washed with PBS and lysed in sodium dodecyl sulfate (SDS) buffer containing Complete Mini Proteinase Inhibitor Cocktail Tablets (Sigma, St. Louis, MO, USA). Protein concentration was determined by Lowry protein assay (Sigma, St. Louis, MO, USA). The protein lysates (30 μg) were loaded onto 12% polyacrylamide SDS gel and and then, the separated proteins were transferred to polyvinylidene difluoride (PVDF) membranes, blocked, and incubated overnight at 4°C with primary antibodies against β-Actin, AKT and phospho-AKT (Ser473) (Cell Signaling Technology). The membranes were washed and then incubated with a secondary antibody for 1 hour at room temperature using an orbital shaker. After washing, the bands were detected using an enhanced chemiluminescence reagent (Western Chemiluminescent HRP Subtract, Merck Millipore).

## Wound healing assay and morphometric analysis

Eight-week-old female C57/BL6 mice were anesthetized and, after shaving the dorsum, two punctures, one on each side of the midline, were made along the dorsal flank with 4-mm disposable punch biopsy instruments. Full-thickness excisional wounds extending through the *panniculus carnosus* were created and the wounded areas were left uncovered. The animals were randomly divided into three groups (4 in every group), depending on the type of medium (BMSC norCM, hypoCM or vehicle control) injected into the injury site. A total of 100 uL of conditioned medium (3.0–3.3 mg/mL) was injected into four diametrically opposed points around the puncture site at 0 and 2 days after procedures. Control mice were similarly treated with PBS. The puncture sites were subsequently photographed at selected time points using a Nikon Coolpix B700 digital camera. Wound surface size was estimated at each time point, expressed as the percentage of original wound size (day 0). After wound closure, mice were euthanized and the wound areas, along with surrounding tissue, were harvested and processed for immunohistochemical (IHC) or immunofluorescence (IF) analysis.

## Immunofluorescence analysis

BMSCs (2.0 x 10$^4$ cell/well) cultured on 13-mm diameter glass coverslips were fixed in 4% paraformaldehyde (PFA) for 15 min at room temperature. Cells were incubated overnight at 4° C with the following primary antibodies diluted in 10% normal goat serum: goat anti-PDGFR (1:50, R&D System), rabbit anti-NG2 (1:200, Millipore) and rabbit anti-Collagen IV (1:200, Abcam). After washing with PBS, samples were incubated with the secondary antibodies (Alexa Fluor-555 anti-goat; Alexa Fluor-488 anti-rabbit, Invitrogen) for 30 min at room temperature.

The skin wound specimens were fixed in 4% PFA for 12 hours at 4° C and then prepared for histological processing according to [31]. Briefly, tissue samples were incubated in 30% sucrose for 24 hours, embedded in OCT, snap-frozen and cryosectioned (50-μm-thick

sections). All frozen sections were washed in PBS (pH 7.4) three times, and then blocked using 5% calf serum in PBS to eliminate nonspecific protein binding. The primary antibody (goat anti-CD31; R&D system) was diluted and incubated overnight at 4° C. After removing any unbound antibodies by washing, the secondary antibody (Alexa Fluor-488 donkey anti-goat; Invitrogen) was added at a concentration of 1:300, and samples were then incubated for 30 min at room temperature. Any unbound antibodies were again removed with PBS. Slides were prepared using mounting medium (Dako) on glass coverslips for fluorescence microscopy analysis (Leika SP8). For histochemical staining, tissue samples were embedded in paraffin, sectioned (5 μm thickness) and stained with hematoxylin and eosin (H&E) in accordance with standard protocols.

## Statistical analysis

Values are expressed as mean +/- standard deviation for at least three independent experiments. The statistical analysis was performed using the Student's t-test or a one way analysis of variance (ANOVA) test with the GraphPad Prism software (Version 6.0). A P value of less than 0.05 was considered statistically significant.

## Results

### Hypoxic preconditioning does not affect the characterization of BMSCs from SCD patients

All experiments were conducted using BMSCs from nine different donors, which were characterized for surface markers and differentiation multipotential, as previously described [24]. Freshly isolated BMSCs from SCD patients showed classic MSC characteristics and displayed spindle-shaped fibroblastic-like morphology when adhered to plastic at P0. Once MSC reached confluence, these were maintained for 48hs in normoxic or hypoxic conditions. Although total cell number increased, no differences were observed between the normoxic and hypoxic cultured cells (Fig 1A). Live-dead staining showed no changes in viability following normoxic or hypoxic preconditioning (Fig 1B).

The expression of characteristic BMSC markers was performed by flow cytometry and immunostaining after normoxic and hypoxic preconditioning. As shown in Fig 2, BMSCs were highly positive for surface markers typical of MSCs, such as CD90, CD105, CD73 and CD29, and were negative for hematopoietic or endothelial markers, such as CD34, CD45 and CD31. Surface marker expression in BMSC cultures after normoxic or hypoxic culturing is shown on S2 Table. Representative markers, such as collagen IV, NG2 and PDGFR-beta, were also detected in the BMSCs (Fig 2B). No significant differences in the expression of any of these markers were observed between the normoxically or hypoxically cultured BMSCs (Fig 2). In sum, hypoxic preconditioning for 48h did not result in any detectable changes with respect to the surface and intracellular markers evaluated.

### Hypoxic preconditioning upregulates expression of trophic factors in the secretome of BMSCs from SCD patients

In order to compare the proangiogenic factor profiles of normoxically and hypoxically cultured BMSCs from SCD patients, the secretome contents were estimated using a human proteome profiler array. Experiments were performed on normoxic (nor) or hypoxic (hypo) conditioned medium (CM) generated after 48 h of serum deprivation, since FBS contains high levels of growth factors. Under normoxic conditions, we observed that classic trophic molecules, such as VEGF-A, MCP-1, IL8, IGFBP-2/3, PTX3 and THBS1, were secreted at relatively high concentrations (Fig

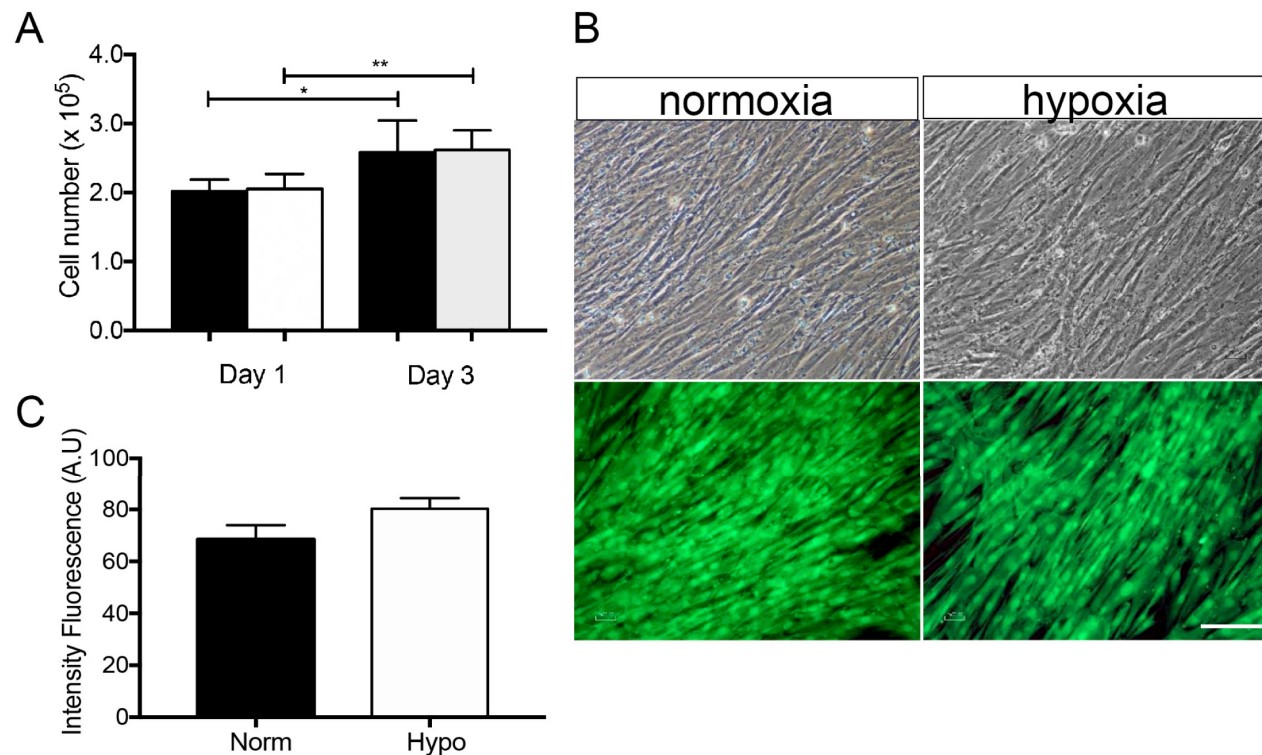

**Fig 1. BMSCs from SCD patients cultured under normoxic and hypoxic conditions showed no significant differences in morphology, expansion and viability. (A)** Isolated BMSCs displaying spindle-shaped fibroblast-like morphology after normoxic and hypoxic preconditioning at passage 3. **(B)** Cell density increased from days 1 to 3, but cell expansion remained unmodified under hypoxic preconditioning. **(C)** Staining with Calcein-AM indicated that BMSC viability remained unaffected by hypoxic preconditioning. Scale bars: 75μm in A. Values are expressed as means ± SD of at least three independent experiments. *p< 0.05; **p< 0.01.

3A). However, the hypoxic secretome was rich in molecules related to angiogenesis and tissue repair processes when compared to the normoxic BMSC secretome (Fig 3B).

Direct analysis of BMSC-CM contents showed that, of all the 55 trophic factors examined, 21 secreted growth factors/cytokines were consistently more abundant in hypoCM than in norCM. The hypoCM molecules with no less than three times higher expression than norCM were grouped into distinct categories: pro-angiogenic factors (angiogenin, angiopoietin-1/2, EG-VEGF, endothelin-1, endostatin, FGF-b, TGF-b1, PIGF and prolactin), tissue repair-related factors (HGF and TIMP-4) and anti-angiogenic factors (platelet factor 4, IGFBP-1 and serpin F1).

The relative expression of transcripts between the normoxic and hypoxic preconditioned BMSCs was evaluated by real time PCR. Of all the 11 genes examined, the transcriptional levels of six genes showed more than a three-fold difference between normoxic (upregulation of THBS) and hypoxic (upregulation of Col-4, FGF-1, HGF, SDF-1a, Il-8) (Fig 3C) conditions. These data indicate that hypoxic preconditioning modulated and enhanced the expression of trophic factors in the secretome of BMSCs from SCD patients.

### The secretome of BMSCs from SCD patients promotes endothelial cell sprouting and migration

HUVEC tube formation and migration induced by norCM or hypoCM were assessed to determine the ability of the secretome of BMSCs from SCD patients to support *in vitro* angiogenesis. In this assay, HUVECs seeded on collagen-coated cytodex beads were embedded in a

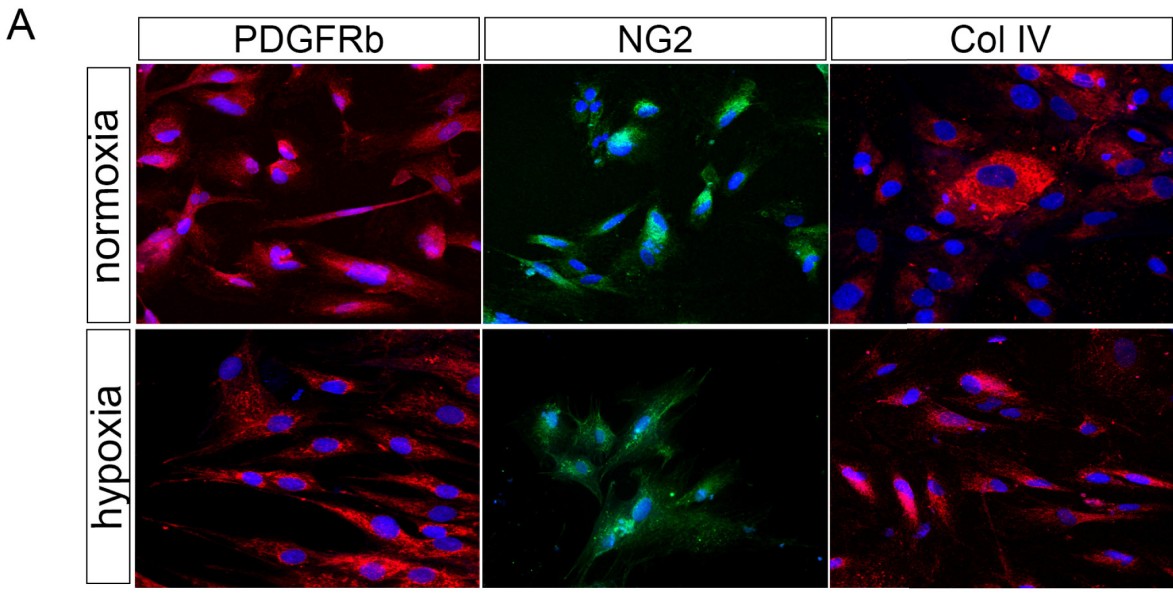

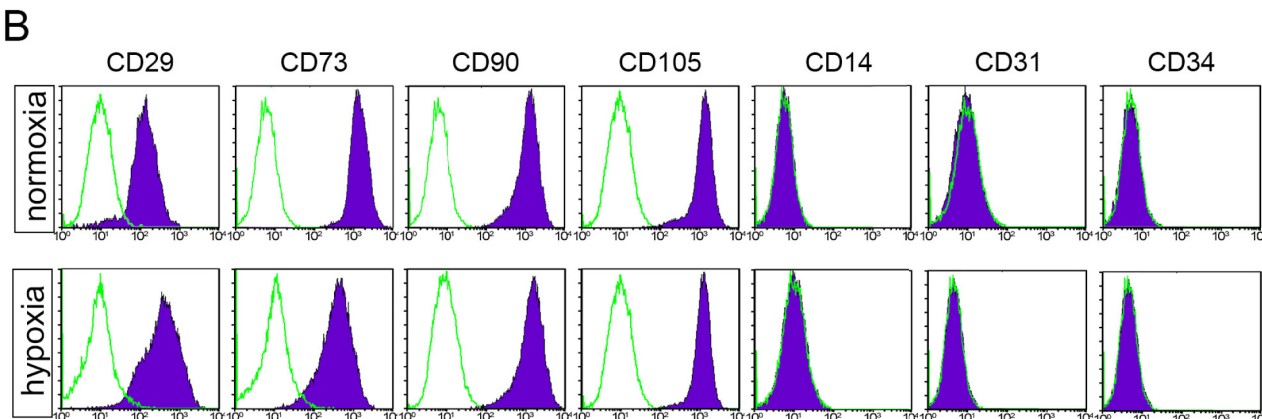

**Fig 2. BMSCs from SCD patients cultured in normoxic and hypoxic conditions display similar immunophenotypes. (A)** Immunocytochemistry detection of NG2, COL4 and PDGFRβ-positive BMSCs after normoxic and hypoxic conditioning. **(B)** Flow cytometric analysis of BMSCs from SCD patients. Specific staining (purple) and the respective isotype-matched control (green line) are shown. Each figure is representative of positive staining in a given cell population. Similar results were observed in eight independent experiments.

3-dimensional fibrin matrix. NorCM or hypoCM were added on top of gel. Fig 4A demonstrates that numbers of sprouts/beads (3.7 ± 1.5 vs. 1.9 ± 0.8 sprout/bead, p<0.01) and sprout lengths (189 ± 120.6 vs. 122.3 ± 88 μm, p<0.05) were significantly increased in the presence of hypoCM as compared to norCM. By contrast, minimal sprouting was observed in the vehicle control medium (Fig 4A). Furthermore, the migration of HUVECs, measured by a monolayer wound assay, was significantly greater in the presence of hypoCM (Fig 4B). HypoCM and norCM conditions significantly facilitated monolayer wound closure in comparison to the vehicle control medium (96.0 ± 7.4, 74.7 ± 11.1 vs 16.3 ± 3.1, p<0.01). These data indicate that hypoxic preconditioning was highly correlated with *in vitro* EC migration and proangiogenic activities in BMSCs from SCD patients.

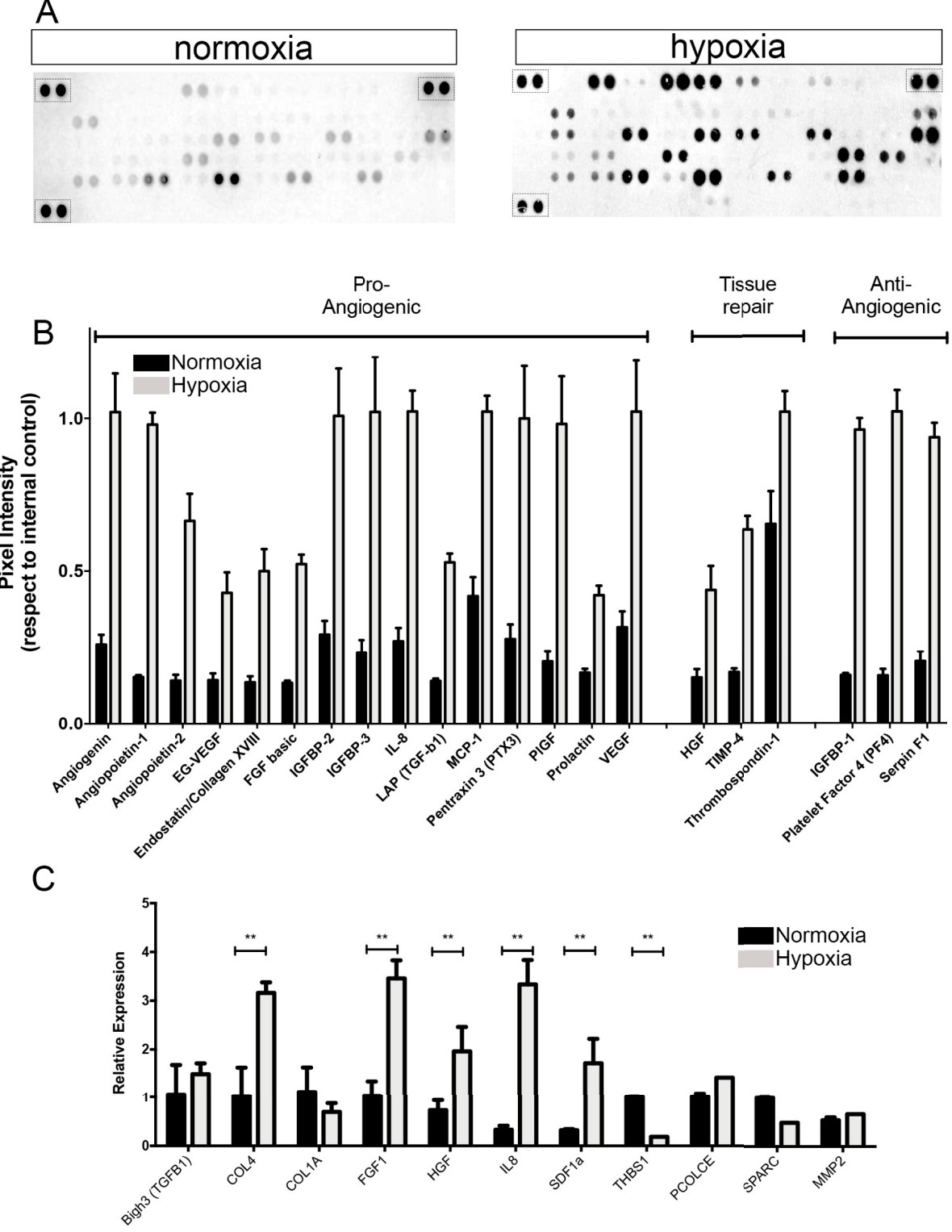

**Fig 3. Hypoxically preconditioned secretome of BMSCs from SCD patients presents enrichment of angiogenic factors. (A)** Differential secretion of trophic factors in conditioned medium during 48h of hypoxic (right) or normoxic (left) culturing. Representative images of membrane-based antibody arrays are shown. Dashed black boxes indicate positive internal control areas. **(B)** Relative expression levels of secreted bioactive factors categorized according to primary function. Quantification of mean spot pixel density was normalized to reference spots. **(C)** Relative expression of genes related to angiogenesis and tissue repair in BMSCs from SCD patients after culturing for 48hs under

normoxic or hypoxic conditions. Total RNA was isolated and expression levels were measured by quantitative real time RT-PCR. Representative data from three independent experiments are shown as means ± SD. Asterisks correspond to statistical analysis comparisons of each condition ($^*p < 0.05$; $^{**}p < 0.01$).

The effects on angiogenesis of HypoCM and norCM were also examined using an *in vivo* assay in C57Bl6 mice. The induction of new blood vessel formation was not evident in the angioreactor containing PBS alone. By contrast, the inclusion of either CM significantly induced the development of new blood vessels. HypoCM was shown to induce greater angiogenesis than norCM, which had less pronounced effect (Fig 4C). These results clearly indicate that hypoxic preconditioning results in *in vivo* pro-angiogenic activity.

### The secretome of BMSCs from SCD patients enhances the proliferation and survival of endothelial cells

To examine the effects of normoxic and hypoxic BMSC secretomes on endothelial cell proliferation, HUVECs were serum-starved for four hours, then stimulated with CM or vehicle control medium for 24 hours. BrdU incorporation in HUVECs was consistently higher in the presence of hypoCM compared to norCM ($p < 0.05$) (Fig 5A). In addition, following serum starvation, hypoCM significantly reduced the number of HUVECs undergoing apoptosis (Fig 5B). Furthermore, phosphorylation of the pro-survival kinase AKT was enhanced under hypoCM ($p < 0.05$), indicating that culturing hypoxic conditions activate survival mechanisms in HUVECs.

### The secretome of BMSCs from SCD patients enhanced skin wound healing in a mouse model

The therapeutic efficacy of hypoCM and norCM was evaluated in a murine excisional wound healing model. Both hypoCM and norCM significantly accelerated wound healing in comparison to controls (Fig 6A). At day 7, a smaller wound area was observed in mice injected with either hypoCM or norCM compared with the vehicle control group. Immunofluorescence analysis showed similar increases in CD31+ (endothelial cell density) and a-SMA (smooth muscle cells) expression in the norCM and hypoCM wounds as compared to controls (Fig 6B). Taken together, these results indicate that norCM and hypoCM wound treatment increased the vascularization of newly formed tissue, an essential component of the wound healing process.

## Discussion

BMSCs are considered to hold strong potential for the development of cell-based therapies designed to enhance angiogenesis during the wound-healing process, and also have beneficial effects on skin regeneration. However, the dysfunction of BMSCs in different bone marrow disorders and pathological conditions has narrowed treatment possibilities [32–34], making the evaluation of these cells' biological functioning crucial prior to therapeutic applications in sickle cell disease patients. The present study provides evidence that the secretome of BMSCs from SCD patients demonstrates pro-angiogenic properties and promotes wound healing. Conditioned media from hypoxic BMSC cultures promoted greater increases in both the proliferation and migration of HUVECs in comparison to cells cultured normoxically or with vehicle control medium. In addition, the introduction of preconditioned medium into skin wounds resulted in a significant improvement in wound closure and the formation of a mature vascular network, confirming that hypoxically preconditioned BMSC media from SCD patients presents considerable proangiogenic and reparative potential.

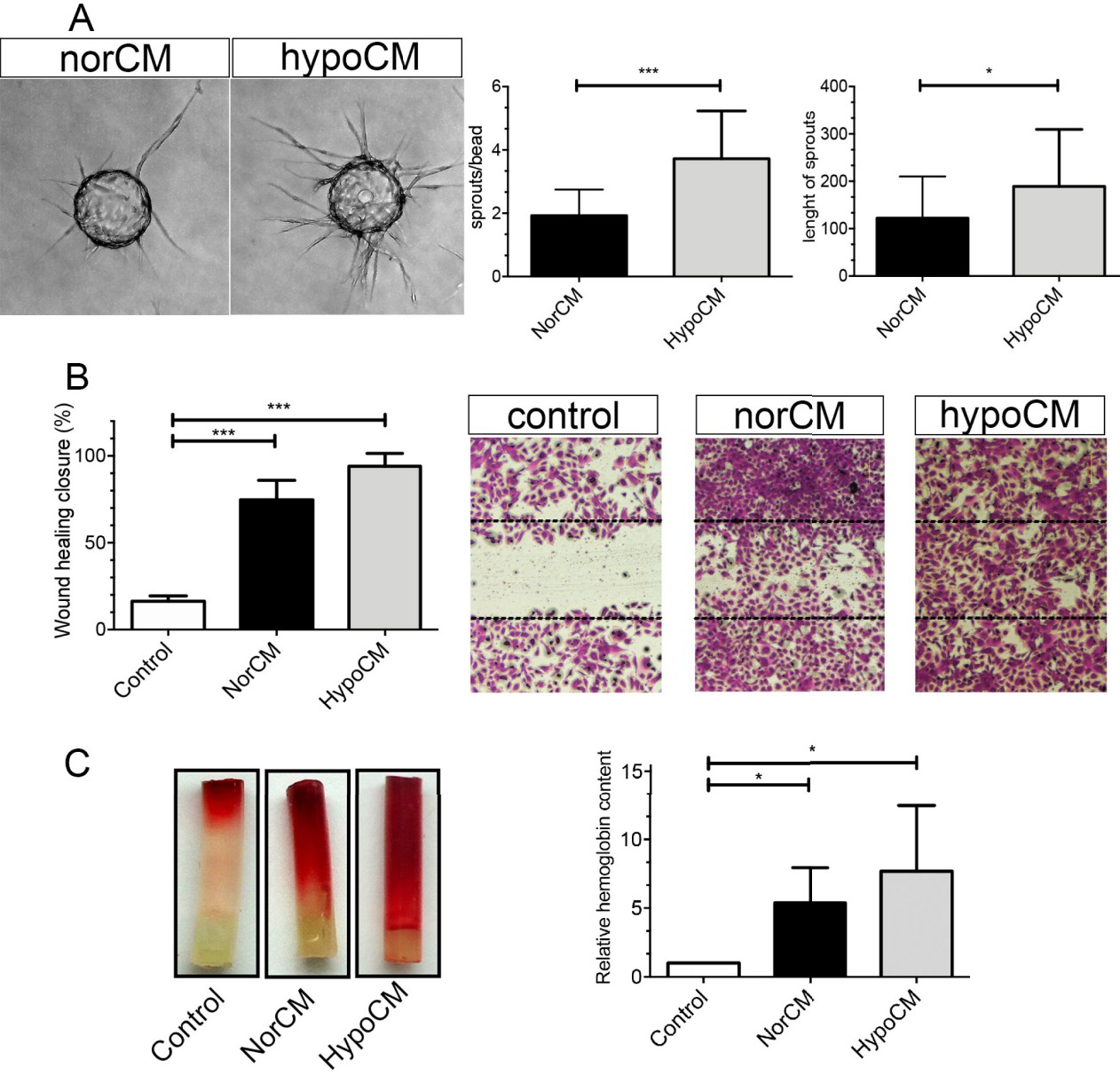

**Fig 4. The secretome of BMSCs from SCD patients promotes *in vitro* endothelial cell migration, sprouting formation and *in vivo* angiogenesis.** (**A**) Representative images of EC-coated beads with norCM or hypoCM overlaid on gel. Quantitation of EC sprouting formation and sprout length in arbitrary units formed after 4 days in fibrin gel overlaid with norCM or hypoCM. (**B**) A representative scratch wound assay made on the HUVEC monolayer. Changes after 24h under ordinal vehicle control medium, norCM, or hypoCM are depicted in the panel. Yellow lines indicate the edge of cell mobilization. Wound healing closure after the initial time point is expressed as the percentage of open area compared to the initial time point for each groups. (**C**) *In vivo* matrigel tube assay. Matrigel-filled angioreactors containing control medium, norCM or hypoCM were subcutaneously implanted into a dorsal area (n = 3) and vessels were allowed to infiltrate. Angioreactors were recovered at 11 days. After visual inspection, hemoglobin content was determined. Values are expressed as means ± SD of at least three independent experiments. *p< 0.05; **p< 0.01. Scale bars: 100 μm in A, B.

Hypoxic preconditioning (i.e., short-term exposure to hypoxia) represents a promising strategy for increasing cellular survival in and the self-renewing capacity of human BMSCs, as well as improving the immunomodulatory and regenerative potential of these cells. However,

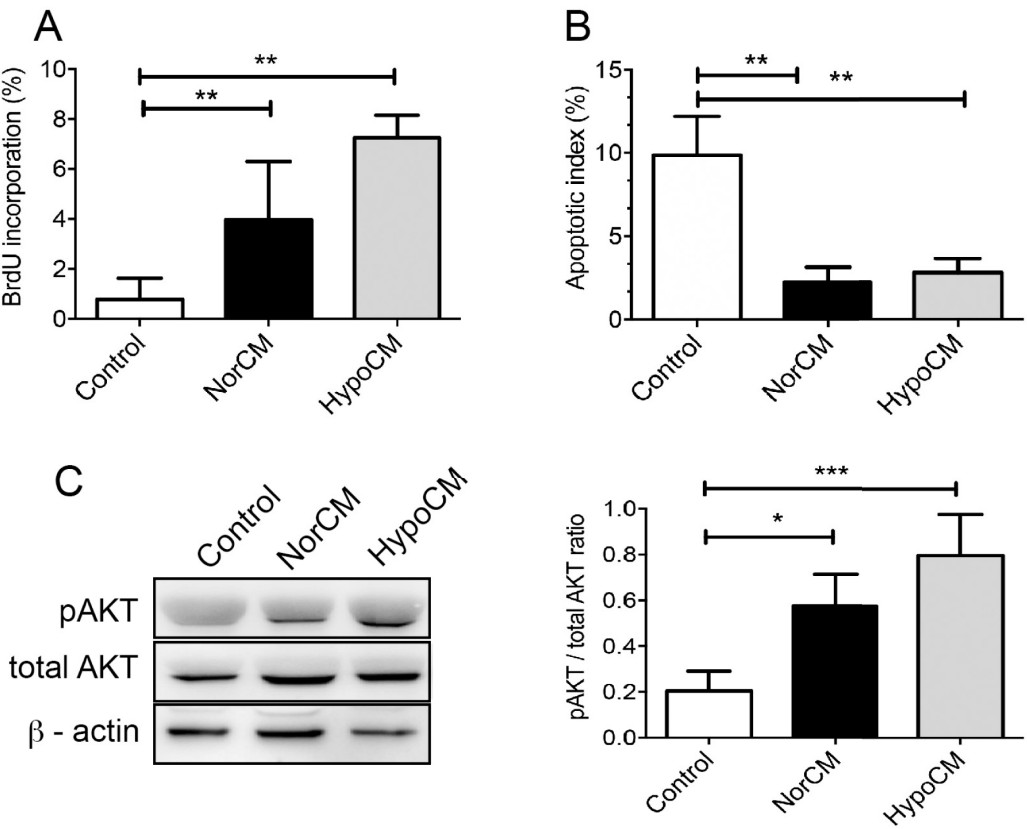

**Fig 5. The hypoCM secretome enhanced HUVEC proliferation and survival compared to the norCM secretome. (A)** BrdU incorporation after 24h in the presence of norCM or hypoCM. HUVECs were pulse-labeled with 10 μM BrdU in culture medium for 4 h before immunocytochemistry detection and analysis. **(B)** HUVECs were cultured for 24h in medium alone (control) or supplemented with either norCM or hypoCM. TUNEL-positive cells were detected and the apoptotic index was calculated as the average number of positive cells compared to the total number of cells in at least six visual fields. Values shown are mean ± SD of at least three independent experiments. **(C)** Western blot analysis of pAKT/AKT signals obtained in HUVECs exposed for 5min to medium alone (control), norCM or hypoCM. HypoCM upregulated expression of pAKT in HUVECs. Results indicate mean normalized expression relative to control ± SD. $^*$p< 0.05; $^{**}$ p<0.01.

the effects on BMSCs produced by hypoxic preconditioning are highly dependent on the culture conditions, cell source and composition of growth medium [15, 35, 36]. Some studies reported that short-term hypoxia preconditioning significantly increased cell viability and induced stem cell properties [17], while others have reported either negative or no effects on BMSCs [37–39]. While hypoxia induces cell cycle arrest in mammalian cells, mesenchymal stem cells are highly resistant to stress conditions [40]. The present study demonstrated that although short-term hypoxic preconditioning had no effect on cell expansion efficacy or viability, the expression and secretion of trophic factors was found to be upregulated in comparison to normoxic culture conditions. Nonetheless, BMSCs subjected to short-term hypoxia exhibited similar cellular marker expression as those cultured under normoxic conditions. These results indicate that the short-term hypoxic preconditioning employed herein did not deleteriously affect cell viability and was capable of inducing a secretory phenotype.

Recent studies have demonstrated the brief survival of BMSCs following implantation, and that the benefits of BMSC therapy could arise from the vast variety of factors secreted by these cells [41–43]. The cell secretome is a collective term defined as the set of secreted trophic

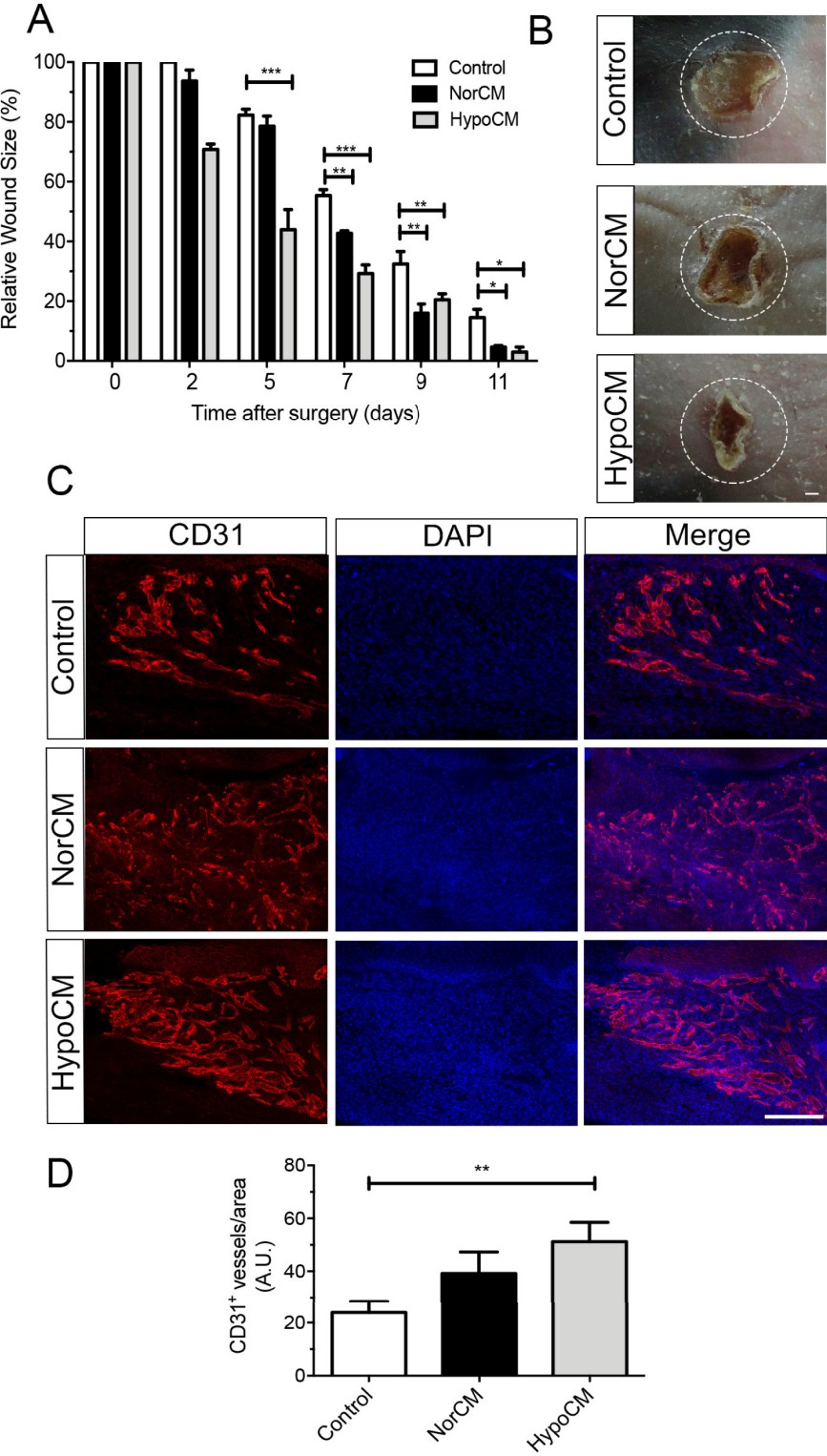

**Fig 6. Hypoxically conditioned medium accelerated wound closure and microvessel density. (A)** The wound area was calculated using images taken on days 2, 5, 7, and 11. **(B)** Representative macroscopic images showing cutaneous wounds on day 7 after injection of control, norCM or hypoCM. The dashed circles indicate the original wound margin. **(C)** Confocal analysis of microvascular density inside wounds. Whole-mount skin immunofluorescence

detection of endothelial cell microvessels (CD31+) at day 7 after wound treatment with control, norCM or hypoCM. **(D)** Microvascular density was quantified as the average number of CD31+ microvessels per viewing field. Values are expressed as means ± SD. *p< 0.05; **p< 0.01; ***p< 0.005. Scale bars: 500 μm in B, 100 μm in C.

factors/soluble molecules (proteins, free nucleic acids, lipids) and extracellular vesicles (exosomes, microvesicles, membrane particles). To date, it has been shown that both of these components may be capable of independently trigger angiogenesis, tissue regeneration and repair [44–46]. Discrimination of which of these components is more effective in producing therapeutic effects is still matter of debate [40, 41, 43]. Here, we were particularly interested in the effects of the whole cell secretome, without fractionation of its sub-component. Preparation of the whole cell-secretome is more economical, more practical, and less time-consuming for clinical application [42]. Recent reports have also described that processing methods to separate soluble components from the microvesicles are labor-intensive, may influence paracrine factor concentration and add several disadvantages that complicate further analysis [47, 48].

The present report found alterations in the secreted factors present in culture medium after hypoxic preconditioning, which may account for the enhanced wound healing and microvascular recovery observed herein. Therefore, we compared the compositions of the secretomes produced in response to normoxic and hypoxic preconditioning using a proteome antibody array. Our results indicated that hypoxic preconditioning led to the increased expression of 17 trophic factors, including four important biomarkers previously implicated in vascular regeneration: angiogenin, interleukin-8, monocyte chemoattractant protein-1 and vascular endothelial growth factor [10]. Some studies have additionally shown that the trophic factors HGF, IGF-1, ang1/2 and bFGF, in addition to promoting angiogenesis, also present anti-inflammatory and neurogenic properties that may either directly or indirectly enhance healing after microvascular or skin injury [42, 49, 50]. In addition, Kim and colleagues described that hypoxia-preconditioning induced BMSCs to express higher levels of HIF-1α and growth factors GDNF, BDNF, VEGF, Ang-1 and SDF-1, as well as its receptor CXCR4, all of which have been linked to neovascularization [51]. These authors additionally found increased expression of EPO and its receptor EPOR, a neuroprotective and pro-angiogenic molecule [51].

The proteome antibody array technique employed herein is limited by the number of specific proteins that can be detected using this approach. Using LC-MS/MS for a more systematic analysis of the human BMSC secretome, Jiang and colleagues identified a diverse range of secretory products associated with microvascular recovery as a result of hypoxic preconditioning [11]. Moreover, an extensive proteomic analysis of the secretome derived from hypoxia-conditioned BMSCs showed that many bioactive factors induced angiogenesis in endothelial cells via the activation of the NFκB pathway [44], while another study implicated the Wnt signaling pathway in the attenuation of vessel injury [45]. While hypoxia seems to be linked to increased secretion of trophic factors in general [52], further research must be carried out to better understand the contributions made by these pathways in the context of SCD. In addition, many potential molecular mechanisms underlying hypoxic CM-mediated functional recovery, which may act as treatment targets in future clinical applications, require additional investigation.

The formation of new blood vessels is a critical step in normal wound healing. Delayed wound healing and chronic wound formation is observed in conditions associated with impaired angiogenesis, such as diabetes and sickle cell disease [2]. The present study demonstrated that the secretome of BMSCs from SCD patients regulates endothelial cell migration, proliferation and survival. These findings were associated with the activation of the PI3k-AKT pathway and the expression of phosphorylated AKT in HUVECs. These data are consistent

with previous reports which showed that the conditioned medium derived from hypoxic BMSCs significantly increased endothelial cell proliferation and migration, thereby promoting early events of angiogenesis [17; 35]. In addition, some authors have described, *in vivo*, the recruitment of fibroblasts and keratinocytes after wound treatment with a hypoxic BMSC secretome [53]. The PI3k-AKT pathway, a classical regulator of cell cycle and survival, has been implicated in ischemic conditions [11]. Moreover, activation of the AKT signaling pathway in endothelial-progenitor cells has been linked to wound healing and microvascular regeneration [54]. These findings suggest that the injection of BMSC secretomes from SCD patients could positively affect the local microenvironment, thusly facilitating the proliferation, recruitment and migration of resident endothelial cells in response to chemoattractants, which may reinforce skin regeneration and repair in conditions associated with impaired angiogenesis, e.g. sickle cell disease.

The therapeutic benefits of the BMSC secretome have been well demonstrated in numerous experimental, pre-clinical and clinical models [41, 42]. Secretome-based approaches may present considerable therapeutic advantages over living cells in terms of facilitating manufacturing, storage, biocompatibility, non-immunogenicity, and non-tumorigenicity. Moreover, secretomes are more physiologically stable than cells, and constitute an efficient biological therapeutic agent. Since the secretome profile varies significantly among BMSC populations derived from different donors and anatomical locations [10, 19, 55], it is important that an appropriate secretome signature be chosen in accordance with a specific therapeutic end point. In this context, additional studies focusing on optimal protein concentrations, administration frequency and optimum injection volume could contribute to the therapeutic success of the BMSC secretome in promoting vascular regeneration and wound healing in individuals with sickle cell disease.

In conclusion, BMSCs subjected to hypoxic preconditioning demonstrated an increased ability to secrete bioactive trophic factors in culture medium, thereby providing paracrine effects to endothelial cells. This finding indicates that the hypoxic secretome of BMSCs from SCD patients presents promising potential to develop regenerative medicine strategies to enhance tissue repair after skin injury.

## Supporting information

**S1 Table. List of primer sequences used for RT-PCR.**
(PDF)

**S2 Table. Flow cytometry analysis of BMSC cultures isolated from SCD patients.** BMSC were isolated, expanded and preconditioned in normoxic or hypoxic conditions for 48hs. No significant differences in surface marker expression were found between normoxia or hypoxia groups. Data represent mean ±SD of three independent experiments.
(PDF)

**S3 Table. Raw data (Hematological characteristics of SCD patients).**
(PDF)

## Acknowledgments

This work was financially supported by the Brazilian Ministry of Health, the Brazilian National Research Council (CNPq) grant 443137/2016-1 to VF, and Research Support Foundation of the State of Bahia (FAPESB). The funders had no role in study design, data collection and analysis, decision to publish, or preparation of the manuscript. The authors would like to thank Andris K. Walter for English language revision and manuscript copyediting assistance.

## Author Contributions

**Conceptualization:** Mari Cleide Sogayar, Vitor Fortuna.

**Formal analysis:** Tiago O. Ribeiro, Brysa M. Silveira, Mercia C. Meira, Ana C. O. Carreira, Roberto Meyer.

**Funding acquisition:** Vitor Fortuna.

**Investigation:** Mari Cleide Sogayar.

**Methodology:** Tiago O. Ribeiro, Brysa M. Silveira, Mercia C. Meira, Ana C. O. Carreira.

**Project administration:** Tiago O. Ribeiro, Brysa M. Silveira.

**Supervision:** Mari Cleide Sogayar, Roberto Meyer, Vitor Fortuna.

**Writing – original draft:** Vitor Fortuna.

**Writing – review & editing:** Vitor Fortuna.

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
