## [Decision Letter · Decision Letter 0]

18 Jun 2019

PONE-D-19-14991

Investigating the potential of the secretome of mesenchymal stem cells derived from sickle cell disease patients

PLOS ONE

Dear Dr Fortuna,

Thank you for submitting your manuscript to PLOS ONE. After careful consideration, we feel that it has merit but does not fully meet PLOS ONE’s publication criteria as it currently stands. Therefore, we invite you to submit a revised version of the manuscript that addresses the points raised during the review process.

We would appreciate receiving your revised manuscript by Aug 02 2019 11:59PM. To enhance the reproducibility of your results, we recommend that if applicable you deposit your laboratory protocols in protocols.io, where a protocol can be assigned its own identifier (DOI) such that it can be cited independently in the future. For instructions see: http://journals.plos.org/plosone/s/submission-guidelines#loc-laboratory-protocols

We look forward to receiving your revised manuscript.

Kind regards,

Carlos E. Ambrósio, Ph.D

Academic Editor

PLOS ONE

Journal Requirements:

Reviewers' comments:

Reviewer's Responses to Questions

**Comments to the Author**

1. Is the manuscript technically sound, and do the data support the conclusions?

Reviewer #1: Yes

Reviewer #2: Yes

2. Has the statistical analysis been performed appropriately and rigorously? 

Reviewer #1: Yes

Reviewer #2: Yes

3. Have the authors made all data underlying the findings in their manuscript fully available?

Reviewer #1: Yes

Reviewer #2: Yes

4. Is the manuscript presented in an intelligible fashion and written in standard English?

Reviewer #1: Yes

Reviewer #2: Yes

5. Review Comments to the Author

Reviewer #1: The Manuscript named “Investigating the potential of the secretome of mesenchymal stem cells derived from sickle cell disease patients” presented is well structured and designed.

I believe that the manuscript has potential to be publish after some clarifications and changes.

Abstract

Why did the authors affirmed that they tested a hypothetically molecules? Why did the authors test an unsure molecules?

Line 25: “in addition to the expression of paracrine molecules that hypothetically contribute to angiogenesis and skin regeneration”

Line 35: Please review this statement: “In sum, culturing under hypoxic conditions produced profound effects on the BMSC secretome, as demonstrated by the expression of trophic paracrine factors involved in angiogenesis and skin regeneration”.

Introduction

Line 59: “and promotes tissue healing and the formation of new blood vessels.”

Change and promotes for “, promote”. And the formation for “formation”

Line 60: the main bioactive factors ghange for “important bioactive factor”

Line 63: Please review this statement: “design highly potent proangiogenic MSC-based cell therapies”. The word design didn’t fit well with the maining.

Line 66- the gap: However, in SCD, 66 the key factors secreted by BMSCs that possess the potential to promote angiogenesis and tissue repair have not been identified to date.

Line 78: Please add the reference for- “The conditioned medium derived from the BMSC culture has been reported to serve multiple positive functions in tissue regeneration”.

Line 82: Please add the references for- “Although numerous studies using BMSCs and their conditioned mediums as potential therapeutic agents have been published”.

Material and Methods

Line 137: “The cells and the CM were then collected and processed at 3200 rpm for 20 min at 4 o C, and kept at -70 o C until use”.

Was 0.5% oxygen for 48h enough to really see a hypoxic result?

Could -70 C affect the results?

When the authors collected CM, was it filtered?

Did the authors remove extracellular vesicles (which are bioactive vesicles) from CM?

Line 150: “Relative mRNA expression of the target genes”

Was these mRNA isolated from CM or BMSC?

Line 190: After a 1-hour incubation period at 37 o C, the angioreactors were subcutaneously implanted into the dorsal flanks of 8-week-old 192 female C57/BL6 mice.

How many animals was used?

Line 281: In “were highly positive for surface” Please add the percentage of each marker.

How the authors did decide the secretome concertation/quantity for the in vivo study?

Results

The results sections are lined with the M & M.

Fig 3B add the STDEV or STD error on the graph.

Discussion

Line 460: In “Recent studies ….”, please add more references to this affirmative.

Other points:

-In general, the authors performed a great study with very interesting results. But it is unclear if they considered in the M & M the proteins contained in the extracellular vesicles as part of the analyzed proteins or not. I suggest to the authors to let clear this point in the M & M.

There as several studies indicating that extracellular vesicles have angiogenesis function:

Pro-Angiogenic Actions of CMC-Derived Extracellular Vesicles Rely on Selective Packaging of Angiopoietin 1 and 2, but Not FGF-2 and VEGF. Wysoczynski M, Pathan A, Moore JB 4th, Farid T, Kim J, Nasr M, Kang Y, Li H, Bolli R.Stem Cell Rev. 2019 May 17. doi: 10.1007/s12015-019-09891-6.

Extracellular Vesicles in Angiogenesis. Todorova D, Simoncini S, Lacroix R, Sabatier F, Dignat-George F. Circ Res. 2017 May 12;120(10):1658-1673. doi: 10.1161/CIRCRESAHA.117.309681.

-In the discussion the authors used the secretome as a general word to describe the compounds from CM, I believe that could be better to use the same concept in the material and methods.

-The authors could discuss better what is in secretome, and why to use a general cell secretion instead of only proteins and/or only extracellular vesicles?

Reviewer #2: In this study everything seems to be according the questions above. The manuscript is presented in standard and correct English, all the experiments appear to have been done appropriately and the conclusion is supported by the data presented.

6. PLOS authors have the option to publish the peer review history of their article (what does this mean?). If published, this will include your full peer review and any attached files.

Reviewer #1: No

Reviewer #2: No

---

## [Author Response · Author response to Decision Letter 0]

23 Jul 2019

Dear Dr. Heber,

 Thank you for sending the comments on our paper, which we have carefully considered. Both reviewers are positive about the paper in principle, but had a number of remaining comments and concerns that we have addressed. We thank the reviewers for their interest and constructive criticism. Briefly, reviewer 1 finds that our results show sufficient evidences to support our conclusions but asked to provide more details about the preparation of our BMSC secretome, to discuss better what a secretome is, and why we should use a general cell secretion instead of only proteins and/or only extracellular vesicles. Therefore, we have included detailed procedures in the Mat&Methods section. We have also rephrased the discussion to reflect more accurately our data. We thank reviewer 1 for pointing out this important topic.

 Additional points raised by the reviewer 1 have been addressed as well, and all changes in the manuscript are highlighted in yellow.

 Reviewer 2 asked us to edit the abstract and text to ensure that our statements are more in line with what the data allow us to conclude. Please see changes in the manuscript in yellow that have been made to address this concern.

 Appended to this letter is our point-by-point response to the comments raised by the reviewers and academic editor. As you notice, we agreed with all the comments raised by the reviewers and editors. We have also revised the paper to conform PLOS ONE guidelines.

 Following the editorial suggestion, we have revised and updated our Data Availability statement: all data underlying the findings are fully available without restriction. All relevant data are within the Supporting Information files.

 We hope that the revised manuscript is accepted for publication in PLOS ONE journal.

 Thanks for your consideration of our revised manuscript.

Best regards,

Vitor Fortuna

---

## [Decision Letter · Decision Letter 1]

22 Aug 2019

Investigating the potential of the secretome of mesenchymal stem cells derived from sickle cell disease patients

PONE-D-19-14991R1

Dear Dr. Vitor Fortuna,

We are pleased to inform you that your manuscript has been judged scientifically suitable for publication and will be formally accepted for publication once it complies with all outstanding technical requirements.

With kind regards,

Carlos E. Ambrósio, Ph.D

Academic Editor

PLOS ONE

Additional Editor Comments (optional):

Reviewers' comments:

Reviewer's Responses to Questions

**Comments to the Author**

1. If the authors have adequately addressed your comments raised in a previous round of review and you feel that this manuscript is now acceptable for publication, you may indicate that here to bypass the “Comments to the Author” section, enter your conflict of interest statement in the “Confidential to Editor” section, and submit your "Accept" recommendation.

Reviewer #1: All comments have been addressed

Reviewer #2: All comments have been addressed

2. Is the manuscript technically sound, and do the data support the conclusions?

Reviewer #1: Yes

Reviewer #2: Yes

3. Has the statistical analysis been performed appropriately and rigorously? 

Reviewer #1: Yes

Reviewer #2: Yes

4. Have the authors made all data underlying the findings in their manuscript fully available?

Reviewer #1: Yes

Reviewer #2: Yes

5. Is the manuscript presented in an intelligible fashion and written in standard English?

Reviewer #1: Yes

Reviewer #2: Yes

6. Review Comments to the Author

Reviewer #1: The authors addressed all suggestions, and I believe that now the manuscript is in a good shape to be published.

Reviewer #2: In this study everything seems to be according the questions above.

The authors corrected all previous questions, the manuscript is presented in standard and correct English, all the experiments appear to have been done appropriately and the conclusion is supported by the data presented.

7. PLOS authors have the option to publish the peer review history of their article (what does this mean?). If published, this will include your full peer review and any attached files.

Reviewer #1: No

Reviewer #2: No

---

## [Editor Report · Acceptance letter]

28 Aug 2019

PONE-D-19-14991R1 

Investigating the potential of the secretome of mesenchymal stem cells derived from sickle cell disease patients 

Dear Dr. Fortuna:

I am pleased to inform you that your manuscript has been deemed suitable for publication in PLOS ONE. Congratulations! Your manuscript is now with our production department. 

With kind regards,

on behalf of

Dr. Carlos E. Ambrósio 

Academic Editor

PLOS ONE